# Integrating Episodic Memory into a Reinforcement Learning Agent using Reservoir Sampling

## Abstract

Episodic memory is a psychology term which refers to the ability to recall specific events from the past. We suggest one advantage of this particular type of memory is the ability to easily assign credit to a specific state when remembered information is found to be useful. Inspired by this idea, and the increasing popularity of external memory mechanisms to handle long-term dependencies in deep learning systems, we propose a novel algorithm which uses a reservoir sampling procedure to maintain an external memory consisting of a fixed number of past states. The algorithm allows a deep reinforcement learning agent to learn online to preferentially remember those states which are found to be useful to recall later on. Critically this method allows for efficient online computation of gradient estimates with respect to the write process of the external memory. Thus unlike most prior mechanisms for external memory it is feasible to use in an online reinforcement learning setting.

Much of reinforcement learning (RL) theory is based on the assumption that the environment has the Markov property, meaning that future states are independent of past states given the present state. This implies the agent has all the information it needs to make an optimal decision at each time and therefore has no need to remember the past. This is however not realistic in general, realistic problems often require significant information from the past to make an informed decision in the present, and there is often no obvious way to incorporate the relevant information into an expanded present state. It is thus desirable to establish techniques for learning a representation of the relevant details of the past (e.g. a memory, or learned state) to facilitate decision making in the present.

A popular approach to integrate information from the past into present decision making is to use some variant of a recurrent neural network, possibly coupled to some form of external memory, trained with backpropagation through time. This can work well for many tasks, but generally requires backpropagating many steps into the past which is not practical in an online RL setting. In purely recurrent architectures one way to make online training practical is to simply truncate gradients after a fixed number of steps. In architectures which include some form of external memory however it is not clear that this is a viable option as the intent of the external memory is generally to capture long term dependencies which would be difficult for a recurrent architecture alone to handle, especially when trained with truncated gradients. Truncating gradients to the external memory would likely greatly hinder this capability.

In this work we explore a method for adding external memory to a reinforcement learning architecture which can be efficiently trained online. We liken our method to the idea of episodic memory from psychology. In this approach the information stored in memory is constrained to consist of a finite set of past states experienced by the agent. In this work, by states we mean observations explicitly provided by the environment. In general, states could be more abstract, such as the internal state of an RNN or predictions generated by something like the Horde architecture of Sutton et al. (2011). By storing states explicitly we enforce that the information recorded also provides the context in which it was recorded. We can therefore assign credit to the recorded state without explicitly backpropagating through time between when the information proves useful and when it was recorded. If a recorded state is found to be useful we train the agent to preferentially remember similar states in the future.

In our approach the set of states in memory at a given time is drawn from a distribution over all $n$-subsets (subsets of size $n$) of visited states, parameterized by a weight value assigned to each state by a trained model. To allow us to draw from such a distribution without maintaining all visited states in memory we introduce a reservoir sampling technique. Reservoir sampling refers to a class of algorithms for sampling from a distribution over $n$-subsets of items from a larger set streamed one item at a time. The goal is to ensure, through specific add and drop probabilities, that the $n$ items in the reservoir at each time-step correspond to a sample from the desired distribution over $n$-subsets of all observed items. Two important examples which sample from different distributions are found in Chao (1982) and Efraimidis & Spirakis (2006). In this work we will define our own distribution and sampling procedure to suit our needs.

# 1 RELATED WORK

Deep learning systems which make use of an external memory have received a lot of interest lately. Two prototypical examples are found in Graves et al. (2014) and the follow-up Graves et al. (2016). These systems use an LSTM controller attached to read and write heads of a fully differentiable external memory and train the combined system to perform algorithmic tasks. Contrary to our approach, training is done entirely by backpropagation through time. See also Zaremba & Sutskever (2015), Joulin & Mikolov (2015), Sukhbaatar et al. (2015), Gulcehre et al. (2017) and Kaiser et al. (2017) for more examples of deep learning systems with integrated external memory.

More directly related to the present work is the application of deep RL to non-markov tasks, in particular Oh et al. (2016). They experiment with architectures using a combination of key-value memory and a recurrent neural networks. The memory saves keys and values corresponding to the last $N$ observations for some integer $N$, thus it is inherently limited in temporal extent but does not require any mechanism for information triage. They test on problems in the Minecraft domain which could provide compelling testbeds for a potential follow-up to the present work. See also Bakker et al. (2003), Wierstra et al. (2010), Zhang et al. (2016) and Hausknecht & Stone (2015) for more examples of applying deep RL to non-markov tasks.

# 2 ARCHITECTURE

Our model is based around an advantage actor critic architecture (Mnih et al., 2016) consisting of separate value and policy networks. In addition we include an external memory $\mathcal{M}$ consisting of a set of $n$ past visited states $(S_{t_0}, .., S_{t_{n-1}})$ with associated importance weights $(w_{t_0}, ..., w_{t_{n-1}})$. The query network $q(S_t)$ outputs a vector of size equal to the state size with tanh activation. At each time step a single item $S_{t_i}$ is drawn from the memory to condition the policy according to:

$$Q(S_{t_i}|\mathcal{M}_t) = \exp\left(\langle q(S_t)|S_{t_i}\rangle/\tau\right) \Big/ \sum_{j=0}^{n-1} \exp\left(\langle q(S_t)|S_{t_j}\rangle/\tau\right) \tag{1}$$

where $\tau$ is a positive learnable temperature parameter. The state, $m_t$, selected from memory is given as input to the policy network along with the current state, both of which condition the resulting policy. Finally the write network takes the current state as input and outputs a single value with sigmoid activation. This value is used to determine how likely the present state is to be written to and subsequently retained in the memory according to the distribution in equation 7 which will be throughly explained in section 3. An illustration of this architecture is shown in figure 1.

# 3 ALGORITHM

For the most part our model is trained using standard stochastic gradient descent on common RL loss functions. The value network is trained by gradient descent on the squared one step temporal different error $\delta_t^2$ where $\delta_t = r_{t+1} + V(S_{t+1}) - V(S_t)$, and the gradient is passed only through $V(S_t)$. The policy is trained using the advantage loss $-\delta_t \log(\pi(a_t|S_t, m_t))$ with gradients passed only through $\pi(a_t|S_t, m_t)$. The query network is trained similarly on the loss $-\delta_t \log(Q(m_t|S_t))$ with gradients passed only through $Q(m_t|S_t)$. We train online, performing one update per time-step with no experience replay. The main innovation of this work is in the training method for the

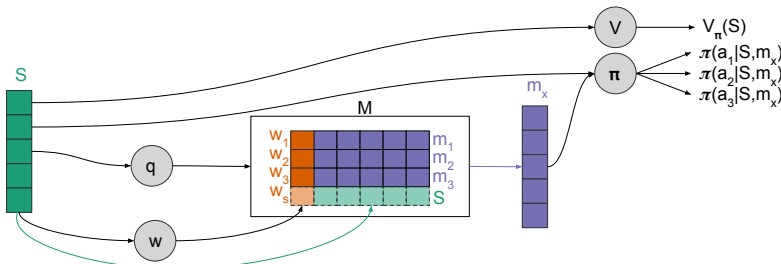

Figure 1: Episodic memory architecture, each grey circle represents a neural network module. Input state (S) is given separately to the query (q), write (w), value (V) and policy ($\pi$) networks at each time step. The query network outputs a vector of size equal to the input state size which is used (via equation 1) to choose a past state from the memory ($m_1$ ,$m_2$ or $m_3$ in the above diagram) to condition the policy. The write network assigns a weight to each new state determining how likely it is to stay in memory. The policy network assigns probabilities to each action conditioned on current state and recalled state. The value network estimates expected return (value) from the current state.

write network which is described in sections 3.1 and 3.2. There are two main desiderata we wish to satisfy with the write network. First we want to use the weights $w(S_t)$ generated by the network in a reservoir sampling algorithm such that the probability of a particular state $S_{\hat{t}}$ being present in memory at any given future time $t > \hat{t}$ is proportional to the associated weight $w(S_{\hat{t}})$. Second we want to obtain estimates of the gradient of the return with respect to the weight of each item in memory such that we can perform approximate gradient descent on the generated weights.

## 3.1 GRADIENT ESTIMATE

For brevity, in this section we will use the notation $E_t[x]$ to denote $E[x|S_0, ..., S_t]$ i.e. the expectation conditioned on the entire history of state visitation up to time $t$. Similarly $P_t(x)$ will represent probability conditioned on the entire history of state visitation. All expectations and probabilities are assumed to be with respect to the current policy, query and write network. Let $\mathcal{A}$ represent the set of available actions and $A_t$ the action selected at time $t$.

### 3.1.1 ONE-STATE MEMORY CASE

To introduce the idea we first present our gradient estimation procedure for the case when our memory can store just one state, and thus there is no need to query. Here $m_t$ represents the state in memory at time $t$ and thus, in the one-state memory case, the state read from memory by the agent at time $t$. Assume the stored memory is drawn from a distribution parameterized as follows by a set of weights $\{w_i | i \in \{0, ..., t-1\}\}$ associated with each state $S_i$ when the state is first visited:

$$P_t(m_t = S_i) = w_i \left/ \sum_{j=0}^{t-1} w_j \right. \tag{2}$$

We can then write the expected return $R_t = r_{t+1} + \gamma r_{t+2} + ...$ as follows:

$$E_t[R_t] = \sum_{k=0}^{t-1} P_t(m_t = S_k) \sum_{a \in \mathcal{A}} \pi(a|S_t, S_k) E_t[R_t|A_t = a] \tag{3}$$

In order to perform gradient descent on the weights $w_i$ we wish to estimate the gradient of this expectation value with respect to each weight. In particular we will derive an estimate of this gradient using an actor critic method which is unbiased if the critic's evaluation is correct. Additionally our estimate will be non-zero only for the $w_i$ associated with the index $i$ such that $m_t = S_i$. This means if our weights $w_i$ are generated by a neural network, we will only have to propagate gradients through the single stored state. This is crucial to allow our algorithm to run online, as otherwise we

would need to store every visited state to compute the gradient estimate.

$$\frac{\partial}{\partial w_i} E_t[R_t] = \sum_{k=0}^{t-1} \left( \frac{\partial P_t(m_t = S_k)}{\partial w_i} \sum_{a \in \mathcal{A}} \pi(a|S_t, S_k) E_t[R_t|A_t = a] \right.$$
$$\left. + P_t(m_t = S_k) \sum_{a \in \mathcal{A}} \pi(a|S_t, S_k) \frac{\partial E_t[R_t|A_t = a]}{\partial w_i} \right) \quad (4)$$

We can rewrite this as:

$$\frac{\partial}{\partial w_i} E_t[R_t] = \frac{1}{w_i} P_t(m_t = S_i) \left( \sum_{a \in \mathcal{A}} \pi(a|S_t, S_i) E_t[R_t|A_t = a] - E_t[R_t] \right)$$
$$+ \gamma E_t \left[ \frac{\partial}{\partial w_i} E_{t+1}[R_{t+1}] \right] \quad (5)$$

See appendix A for a detailed derivation of this expression. We will use a policy gradient approach, similar to REINFORCE (Williams, 1992), to estimate the this gradient using an estimator $G_{i,t}$ such that $E_t[\sum_{\hat{t} \geq t} \gamma^{\hat{t}-t} G_{i,\hat{t}}] \approx \frac{\partial E_t[R_t]}{\partial w_i}$, thus the second term is estimated recursively on subsequent time-steps. In the present work we will focus on the undiscounted episodic case with the start-state value objective, for which it suffices to follow the first term in the above gradient expression for each visited state. This is also true in the continuing case with an average-reward objective. See Sutton et al. (2000) for further discussion of this distinction. Consider the gradient estimator:

$$G_{i,t} = \begin{cases} \delta_t/w_i & \text{if } m_t = S_i \\ 0 & \text{otherwise} \end{cases} \quad (6)$$

which has expectation:

$$E_t[G_{i,t}] = \frac{1}{w_i} P_t(m_t = S_i) \sum_{a \in \mathcal{A}} \pi(a|S_t, S_i) E_t[r_{t+1} + \gamma V(S_{t+1}) - V(S_t)|A_t = a]$$
$$\approx \frac{1}{w_i} P_t(m_t = S_i) \sum_{a \in \mathcal{A}} \pi(a|S_t, S_i)(E_t[r_{t+1} + \gamma E_{t+1}[R_{t+1}]|A_t = a] - E_t[R_t])$$
$$= \frac{1}{w_i} P_t(m_t = S_i) \left( \sum_{a \in \mathcal{A}} \pi(a|S_t, S_i) E_t[R_t|A_t = a] - E_t[R_t] \right)$$

Where the approximation is limited by the accuracy of our value function. In conventional policy gradient subtracting the state value (e.g. using $\delta_t = r_{t+1} + \gamma V(S_{t+1}) - V(S_t)$ instead of $r_{t+1} + \gamma V(S_{t+1})$) is a means of variance reduction. Here it is critical to avoid computing gradients with respect to the denominator of equation 2, which allows our algorithm to run online while computing the gradient with respect to only the weight stored in memory.

Given these estimated gradients with respect to $w_i$ we apply the chain rule to compute $\frac{\partial R_t}{\partial \theta_w} = \frac{\partial R_t}{\partial w_i} \frac{\partial w_i}{\partial \theta_w} \approx \sum_{\hat{t} \geq t} G_{i,\hat{t}} \frac{\partial w(S_i)}{\partial \theta_w}$, for each parameter $\theta_w$ of the write network. This gradient estimate is used in a gradient descent procedure to emphasize retention of states which improve the return.

Gradient estimates are generated based on the stored values of $w_i$ in memory but applied to the parameters of the network at the present time. With online updating, this introduces a potential multiple timescale issue which we conjecture will vanish in the limit of small learning rate, but leave further investigation to future work.

There are a number of ways to extend the distribution defined in equation 2 to the case where multiple elements of a set must be selected (see for example Efraimidis & Spirakis (2006)). We will focus on a generalization which is less explored but which we will see in the following section results in gradient estimates which are an elegant generalization of the single-state memory case.

### 3.1.2 MULTIPLE-STATE MEMORY CASE

In this section and those that follow we will routinely use the notation $\binom{Z}{n}$ where $Z$ is a set and $n$ an integer to indicate the set of all $n$-subsets of $Z$. Note that $\binom{Z}{0} = \{\emptyset\}$ and we adopt the convention $\prod_{x \in \emptyset} x = 1$ thus $\sum_{\hat{Z} \in \binom{Z}{0}} \prod_{x \in \hat{Z}} x = 1$ which will be important in a few places in what follows.

We will introduce some notation to facilitate reasoning about sets of states. Let $T_t = \{t' : 0 \leq t' \leq t - 1\}$ be the set of all time indices from 0 to $t - 1$. Let $\hat{T} \in \binom{T_t}{n}$ be a set of $n$ indices chosen from $T_t$ where $n$ is the memory size. Let $S_{\hat{T}}$ be the set of states $\{S_{\hat{t}} : \hat{t} \in \hat{T}\}$. Let $\mathcal{M}_t$ be the set of states in memory at time $t$. Let $Q(S_{\hat{t}}|S_{\hat{T}})$ be the probability of querying $S_{\hat{t}}$ given $\mathcal{M}_t = S_{\hat{T}}$. The probability for a particular set of states being contained in memory is defined to be the following:

$$P_t(\mathcal{M}_t = S_{\hat{T}}) = \prod_{i \in \hat{T}} w_i \left/ \sum_{\tilde{T} \in \binom{T_t}{n}} \prod_{i \in \tilde{T}} w_i \right. \tag{7}$$

A straightforward extension of the derivation of the equation 5 shows that this choice results in a gradient estimate which is an elegant extension of the one-state memory case. The derivation is given in appendix B, the result for $\frac{\partial}{\partial w_i} E_t[R_t]$ is:

$$\frac{\partial}{\partial w_i} E_t[R_t] = \sum_{\hat{T} \in \left(\binom{T_t}{n} \ni i\right)} \frac{1}{w_i} P_t(\mathcal{M}_t = S_{\hat{T}}) \left( \sum_{j \in \hat{T}} Q(S_j|S_{\hat{T}}) \sum_{a \in \mathcal{A}} \pi(a|S_t, S_j) E_t[R_t|A_t = a] \right.$$
$$\left. - E_t[R_t] \right) + \gamma E_t \left[ \frac{\partial}{\partial w_i} E_{t+1}[R_{t+1}] \right] \tag{8}$$

As in the single-state memory case we recursively handle the second term. To estimate the first term we could choose the following estimator $G_{i,t}$:

$$G_{i,t} = \begin{cases} \delta_t/w_i & \text{if } S_i \in \mathcal{M}_t \\ 0 & \text{otherwise} \end{cases} \tag{9}$$

This estimator is unbiased under the assumption the critic is perfect, however it scales poorly in terms of both variance and computation time as the memory size increases. This is because it requires updating every state in memory regardless of whether it was queried, spreading credit assignment and requiring compute time proportional to the product of the number of states in memory with the number of parameters in the write network. Instead we will further approximate the second term and perform an update only for the queried item. We rewrite the first term of equation 8 as follows:

$$\frac{1}{w_i} P_t(\mathcal{M}_t \ni S_i) \left( P_t(m_t = S_i | \mathcal{M}_t \ni S_i) \left( \sum_{a \in \mathcal{A}} \pi(a|S_t, S_j) E_t[R_t|A_t = a] - E_t[R_t] \right) \right.$$
$$\left. + P_t(m_t \neq S_i | \mathcal{M}_t \ni S_i) \left( \sum_{a \in \mathcal{A}} P_t(A_t = a | \mathcal{M}_t \ni S_i, m_t \neq S_i) E_t[R_t|A_t = a] - E_t[R_t] \right) \right)$$
$$\approx \frac{1}{w_i} P_t(m_t = S_i) \left( \sum_{a \in \mathcal{A}} \pi(a|S_t, S_j) E_t[R_t|A_t = a] - E_t[R_t] \right)$$

This approximation is accurate to the extent that our query network is able to accurately select useful states. To see this, note that if querying a state when it's in memory helps to generate a better expected return a well trained query network should do it with high probability and hence $P_t(m_t \neq S_i | \mathcal{M}_t \ni S_i)$ will be low. On the other hand if querying a state in memory is unhelpful $\left( \sum_{a \in \mathcal{A}} P_t(A_t = a | \mathcal{M}_t \ni S_i, m_t \neq S_i) E_t[R_t|A_t = a] - E_t[R_t] \right)$ will generally be small. With this approximation the gradient estimate becomes identical to the one-state memory case:

$$G_{i,t} = \begin{cases} \delta_t/w_i & \text{if } m_t = S_i \\ 0 & \text{otherwise} \end{cases} \tag{10}$$

While this justification is not rigorous, this approximation should significantly improve computational and sample efficiency, and is used in our experiments in section 4.

## 3.2 RESERVOIR SAMPLING PROCEDURE

---

**Algorithm 1** A reservoir sampling algorithm for drawing samples from equation 11

---

1: $\Omega \leftarrow$ Zeros(n+1)
2: $\tilde{\Omega} \leftarrow$ Zeros(n)
3: $W \leftarrow$ Zeros(n)
4: $\hat{T} \leftarrow$ Zeros(n)
5: **for** time $0 \leq t \leq n-1$ **do**
6:      Receive $w_t$
7:      $W[t] \leftarrow w_t$
8:      $\hat{T}[t] \leftarrow t$
9: **end for**
10: Apply equivalent random permutation to $W$
     and $\hat{T}$
11: $\Omega[n] \leftarrow 1$
12: $\tilde{\Omega}[n-1] \leftarrow 1$
13: **for** $n-1 \geq i \geq 0$ **do**
14:      $\Omega[i] = W[i] \cdot \Omega[i+1]$
15: **end for**
16: **for** $n-2 \geq i \geq 0$ **do**
17:      $\tilde{\Omega}[i] = \Omega[i+1] + W[i] \cdot \tilde{\Omega}[i+1]$
18: **end for**
19: **for all** time $t \geq n$ **do**
20:      Receive $w_t$
21:      UPDATE($w_t$,t)
22: **end for**

1: **function** UPDATE($w$,t)
2:      $\omega \leftarrow w$
3:      $\tau \leftarrow t$
4:      **for** $0 \leq i \leq n-1$ **do**
5:          $\Omega' \leftarrow \Omega[i] + \omega \cdot \tilde{\Omega}[i]$
6:          **if** i=n-1 **then**
7:              $\Omega'' \leftarrow \Omega[i+1]$
8:          **else**
9:              $\Omega'' \leftarrow \Omega[i+1] + \omega \cdot \tilde{\Omega}[i+1]$
10:          **end if**
11:          $P \leftarrow 1 - \frac{\Omega'' \Omega[i]}{\Omega' \Omega[i+1]}$
12:          Swap $\omega$ with $W[i]$ and $\tau$ with $\hat{T}[i]$
     with probability P
13:          $\Omega[i] \leftarrow \Omega'$
14:      **end for**
15:      **for** $n-2 \geq i \geq 0$ **do**
16:          $\tilde{\Omega}[i] = \Omega[i+1] + W[i] \cdot \tilde{\Omega}[i+1]$
17:      **end for**
18: **end function**

---

In the previous section we derived a gradient estimator for our desired memory distribution. in this section we introduce a method for sampling from this distribution online. Specifically we will formulate a reservoir sampling algorithm for drawing a subset $\hat{T}$ of size $n$ from a set of indices $T = \{0, ..., t-1\}$ according to a distribution parameterized by a weight $w_i$ for each index $i \in T$. Following equation 7 the probability for a given subset $\hat{T}$ is defined as follows:

$$\tilde{P}(\hat{T}; T, n) = \prod_{i \in \hat{T}} w_i \bigg/ \sum_{\tilde{T} \in \binom{T}{n}} \prod_{i \in \tilde{T}} w_i \tag{11}$$

This distribution can be sampled from by selecting members sequentially for $i \in \{0, .., n-1\}$ with the following conditional probabilities:

$$\hat{P}(\hat{T}[i] | \hat{T}[0:i-1]; T, n) = w_{\hat{T}[i]} \sum_{\tilde{T} \in \binom{T \setminus \hat{T}[0:i]}{n-i-1}} \prod_{j \in \tilde{T}} w_j \bigg/ \left( (n-i) \sum_{\tilde{T} \in \binom{T \setminus \hat{T}[0:i-1]}{n-i}} \prod_{j \in \tilde{T}} w_j \right) \tag{12}$$

We abuse notation slightly and use $\hat{T}$ to refer to both an ordered vector and the set of its elements.

**Lemma 1.** *Selecting elements sequentially according to equation 12 will result in a vector $\hat{T}$ whose elements correspond to a sample drawn from equation 11.*

*Proof.* See appendix C. □

We use lemma 1 to derive a reservoir sampling procedure which works online to update the reservoir $\hat{T}$ at each time-step when a new index is added to $T$ along with an associated weight. The result

is algorithm 1. At each time-step UPDATE moves through $\hat{T}$ starting from index 0 and chooses whether to swap the item and weight at each index with the ones currently contained in a buffer ($\tau$ and $\omega$, initially set to contain the newly added item and associated weight). The probability of swapping is chosen such that it corrects the conditional probability of the item at each index (conditioned on the items before it) to compensate for the item in the buffer being added to the set of possible items for that index. After doing this sequentially at each index the overall probability of $\hat{T}$ will be correct with the newly added item. Computing the necessary swap probabilities is nontrivial in itself, however we show that it is possible to do this in $O(n)$ time per time-step (where here n is the memory size) by iteratively updating two vectors $\Omega$ and $\tilde{\Omega}$.

**Theorem 1.** *In algorithm 1 let $t$ refer to the parameter of the call to UPDATE, $\hat{T}_t[i]$ refer to the value of $\hat{T}[i]$ when that call is made, and $T_t = \{t' : 0 \leq t' \leq t - 1\}$ refer to the set of all time indices from 0 to $t - 1$. $\forall t \geq n$, $0 \leq i \leq n - 1$:*

$$P(\hat{T}_t[i] = t_i | \hat{T}_t[0 : i - 1] = [t_0, ..., t_{i-1}]) = \hat{P}(t_i | [t_0, ..., t_{i-1}]; T_t, n)$$

*where $[t_0, ..., t_i]$ is any arbitrary vector of unique elements of $T_t$.*

*Proof.* See appendix D. □

**Corollary 1.1.** *At the call to UPDATE with parameter $t$, $\forall t \geq n$, $\hat{T} \in \binom{T_t}{n}$ :*

$$P(\{\hat{T}_t[0], ..., \hat{T}_t[n - 1]\} = \hat{T}) = \tilde{P}(\hat{T}; T_t, n)$$

*Proof.* The proof follows from theorem 1 and lemma 1. □

Corollary 1.1 tells us that for any given time-step algorithm 1 produces reservoirs which are a valid sample from the distribution of equation 11. Note that algorithm 1 runs in $O(n)$ time per time-step where $n$ is the size of the memory. We use this algorithm along with the weights generated by our write network to manage updating the memory on each new state visitation.

The careful reader will notice that with the use of reservoir sampling equation 7 no longer holds explicitly. This is because certain parts of the history may strongly correlate with certain states being in memory at a particular time in the past, which under reservoir sampling will effect the distribution of the present memory. We do not account for this in this work and simply assume for the purpose of estimating gradients that the history of state visitation arises independently of the history of memory content. Further analysis of the implications of this assumption, and whether it can be weakened is left to future work.

## 4 EXPERIMENTS AND RESULTS

We test our algorithm on a toy problem we call "the secret informant problem". The problem is intended to highlight the kinds of sharp, long-term dependencies that are often difficult for recurrent models. The problem is such that in order to behave optimally an agent must remember specific, initially unknown past states. An instance of the problem is shown in figure 2 and a detailed explanation of the problem structure is available in the associated caption. In each training episode a new random instance of the problem is created (with the chain length, number of actions and number of decisions held fixed for a particular training run). This consists of randomly choosing the rewarding action sequence, the location of the informative state for each decision, and the implied action and decision state for each of the uninformative states.

All experiments with our episodic memory architecture are run for 3 repetitions with error bars indicating standard error in the mean over these 3 runs. The architecture and hyper-parameters used in each experiment are identical. We use the architecture from section 2 with 1 hidden layer for the value, query and write networks and 2 hidden layers for policy. The value network and query network outputs each use tanh activation, the policy uses softmax, and the write network uses sigmoid. Each hidden layer has 10 units. We train using ordinary stochastic gradient descent with learning rate 0.005 and gradient estimates generated as specified in section 3.

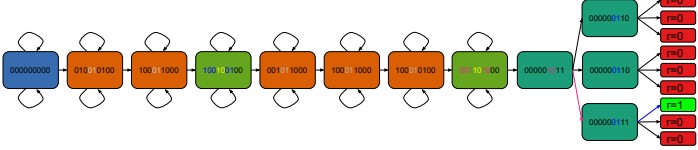

(a) An instance of the secret informant problem

(b) A state of the secret informant problem

Figure 2: (a) shows an instance of the secret informant problem with 3 actions ($\mathcal{A} = \{up, forward, down\}$) and 2 decision states. The **start state** uniquely contains all zeros. In the final 2 states of an episode (which we call **decision states**), the agent must select the right sequence of actions to receive a $+1$ reward, any other action sequence gives reward $0$. At all other states the forward action leads forward along the chain while other actions keep the agent in the same state. The correct action (i.e. the one that leads towards the reward) at each decision state is indicated by a one hot encoding on bits 1-3 of a certain informative state where the pattern in bits 6 and 7 matches those of the decision state itself. **Informative states** are distinguished from **uninformative states** by bits 4 and 5. To succeed the agent must learn to remember informative states with the pattern 10 in bits 4 and 5 and subsequently query them at the associated decision state. (b) shows a particular state of the problem. The first 3 bits are **action indicators**, a one-hot encoding of the action the state is suggesting should be taken at the associated decision state. The next two bits are **informative and uninformative indicators**. If these bits are 01 the state is uninformative, meaning the decision state and associated action it suggests are uniformly random and give no indications of a correct action. If these bits are 10 then the state is informative and the associated action it suggests is on the path toward the reward at the decision state it indicates. The next two bits are **decision state identifiers**, in an informative state they indicate it provides information about the decision state with matching identifier, in a decision state they serve as an identifier for that decision state. Thus, the correct thing for an agent to do in each decision state is to take the action suggested by the informative state with matching values of bits 6 and 7. The next bit is a **decision state indicator** and will be 1 if and only if the state is a decision state. The final bit is a **correct path indicator** and indicates for a decision state whether all the decisions made so far have been correct. This is necessary for our current system because without it the final decision states all look the same and it is not possible to learn via one step updates which decision is correct at the first decision state, in future work we would like to investigate eliminating the need for information like this by using multi-step updates or eligibility traces. The particular state shown above is informative, it indicates that the correct action for the second decision state will be the up action. Versions of this problem can be created with variable length (which we use to refer to the number of informative states plus the number of uninformative states), number of actions, and number of decision states by modifying the above description in the obvious way.

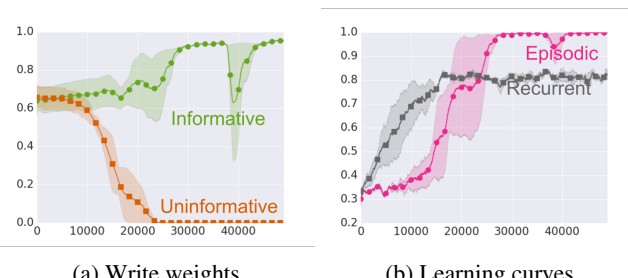

(a) Write weights

(b) Learning curves

Figure 3: Experiment with environment length 10, 1 decision and a 1 state memory. In the 1 state memory case the query module is unnecessary. (a) shows average write weight assigned to informative states (●) and uninformative states (■). (b) shows average return with episodic memory learner (●) and recurrent baseline (■).

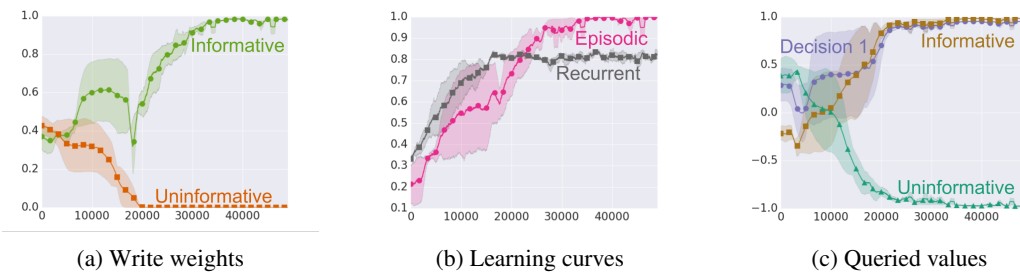

(a) Write weights     (b) Learning curves     (c) Queried values

Figure 4: Experiment with environment length 10, 1 decision and a 3 state memory. In this case the query module is necessarily. (a) shows average write weight assigned to informative states (●) and uninformative states (■). (b) shows average return with episodic memory learner (●) and recurrent baseline (■). (c) shows the value of several relevant query vector elements in the decision state: the uninformative indicator (▲), the informative indicator (■) and the first decision state identifier (●) (unnecessary here since there is only one decision state, but included for uniformity).

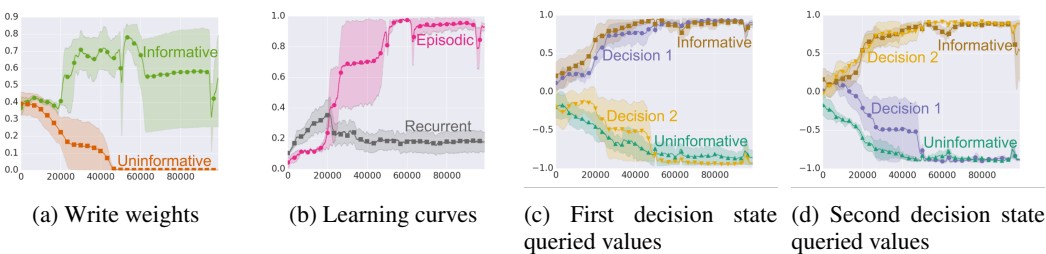

(a) Write weights     (b) Learning curves     (c) First decision state queried values     (d) Second decision state queried values

Figure 5: Experiment with environment length 10, 2 decisions and a 3 state memory. (a) shows average write weight assigned to informative states (●) and uninformative states (■). (b) shows average return with episodic memory learner (●) and recurrent baseline (■). (c) shows the value of several relevant query vector elements in the first decision state: the uninformative indicator (▲), the informative indicator (■), the first decision state identifier (●), and the second decision state identifier (▼). (d) shows the same thing but for the query generated in the second decision state.

We also ran a recurrent baseline with full backpropagation through time which we found required more fine-tuning to train with online updates. To make comparison as meaningful as possible the recurrent baseline used essentially the same architecture but with the entire memory module replaced by a basic GRU (Cho et al., 2014) network of 10 units. Stochastic gradient descent alone was found to give very poor results with the recurrent learner so RMSProp was used instead. Additionally to obtain reasonable results with the recurrent learner, and avoid catastrophic looping behavior, it was necessary to add a discount factor ($\gamma = 0.9$, applied only for learning purposes and not used in computing the plotted return) as well as entropy regularization on the policy with a relatively low weight of 0.0005. Perhaps surprisingly neither of these were necessary with the episodic memory based system as it tended to proceed quickly through the chain without significant looping even without discounting or entropy regularization. We tuned the learning rate and layer width (including the number of recurrent units) for each of the 2 environments on which a recurrent baseline was trained according to highest average performance over the last 100 episodes of a single training run of 25000 episodes for the 1 decision environment, and 50000 episodes for the 2 decision environment. In each case learning rate was selected from $\{0.05 \cdot 2^{-x} : x \in \{0, ..., 9\}\}$ and layer width was selected from $\{5, 10, 15, 20\}$. For the 1 decision environment we ran the recurrent baseline for 3 repeats, for the 2 decision environment due to higher variance we ran it for 10 repeats. This baseline is not intended to be representative of the performance of all possible architectures based on RNN variants trained with backpropagation through time, but merely to provide context for the main experimental results of this work.

Results and descriptions of the experiments are shown in figures 3, 4 and 5, in each plot the x-axis shows number of training episodes. One additional experiment with twice the environment length is shown in appendix E. Notice that in each case the episodic memory learner was is able to learn

a good query policy, drive the write weight of the uninformative states to near 0 while keeping the value for informative states much larger, and obtain close to perfect average return. Comparing figures 3 and 4 it appears that the addition of redundant memory size may accelerate initial learning though it has little effect on the overall convergence time. Comparing figures 4 and 5 the number of episodes to converge appears to roughly double from approximately $25,000$ to $50,000$ with the addition of the extra decision state but the training remains quite stable.

## 5    CONCLUSION

We present a novel algorithm for integrating a form of external memory with trainable reading and writing into a RL agent. The method depends on the observation that if we restrict the information stored in memory to be a set of past visited states, the information recorded also provides the context in which it was recorded. This means it is possible to assign credit to useful information without needing to backpropagate through time to when it was recorded. To achieve this we devise a reservoir sampling technique which uses a sampling procedure we introduce to generate a distribution over memory configurations for which we can derive gradient estimates. The whole algorithm is $O(n)$ in both the number of trainable parameters and the size of the memory. In particular neither memory required nor computation time increase with history length, making it feasible to run in an online RL setting. We show that the resulting algorithm is able to achieve good performance on a toy problem we introduce designed to have sharp long-term dependencies which can be problematic for recurrent models.

ACKNOWLEDGEMENTS

We acknowledge the support of the Natural Sciences and Engineering Council of Canada (NSERC).

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

## A    DERIVATION OF EXPRESSION FOR GRADIENT FOR ONE-STATE MEMORY

$$P_t(m_t = S_i) = w_i \left/ \sum_{j=0}^{t-1} w_j \right.$$

$$E_t[R_t] = \sum_{k=0}^{t-1} P_t(m_t = S_k) \sum_{a \in \mathcal{A}} \pi(a|S_t, S_k) E_t[R_t|A_t = a]$$

$$\frac{\partial}{\partial w_i} E_t[R_t] = \sum_{k=0}^{t-1} \left( \frac{\partial P_t(m_t = S_k)}{\partial w_i} \sum_{a \in \mathcal{A}} \pi(a|S_t, S_k) E_t[R_t|A_t = a] \right.$$
$$\left. + P_t(m_t = S_k) \sum_{a \in \mathcal{A}} \pi(a|S_t, S_k) \frac{\partial E_t[R_t|A_t = a]}{\partial w_i} \right)$$

Working out the first term:

$$\sum_{k=0}^{t-1} \frac{\partial P_t(m_t = S_k)}{\partial w_i} \sum_{a \in \mathcal{A}} \pi(a|S_t, S_k) E_t[R_t|A_t = a]$$

$$= \sum_{k=0}^{t-1} \frac{\partial \log(P_t(m_t = S_k))}{\partial w_i} P_t(m_t = S_k) \sum_{a \in \mathcal{A}} \pi(a|S_t, S_k) E_t[R_t|A_t = a]$$

$$= \sum_{k=0}^{t-1} \frac{\partial}{\partial w_i}(\log(w_k) - \log(\sum_{j=0}^{t-1} w_j)) P_t(m_t = S_k) \sum_{a \in \mathcal{A}} \pi(a|S_t, S_k) E_t[R_t|A_t = a]$$

$$= \frac{1}{w_i} P_t(m_t = S_i) \sum_{a \in \mathcal{A}} \pi(a|S_t, S_i) E_t[R_t|A_t = a]$$

$$- \sum_{k=0}^{t-1} \frac{1}{\sum_j w_j} P_t(m_t = S_k) \sum_{a \in \mathcal{A}} \pi(a|S_t, S_k) E_t[R_t|A_t = a]$$

$$= \frac{1}{w_i} P_t(m_t = S_i) \sum_{a \in \mathcal{A}} \pi(a|S_t, S_i) E_t[R_t|A_t = a] - \frac{1}{\sum_j w_j} E_t[R_t]$$

$$= \frac{1}{w_i} P_t(m_t = S_i) \left( \sum_{a \in \mathcal{A}} \pi(a|S_t, S_i) E_t[R_t|A_t = a] - E_t[R_t] \right)$$

Working out the second term:

$$\sum_{k=0}^{t-1} P_t(m_t = S_k) \sum_{a \in \mathcal{A}} \pi(a|S_t, S_k) \frac{\partial E_t[R_t|A_t = a]}{\partial w_i}$$

$$= \sum_{k=0}^{t-1} P_t(m_t = S_k) \sum_{a \in \mathcal{A}} \pi(a|S_t, S_k) \left( \frac{\partial E_t[r_{t+1}|A_t = a]}{\partial w_i} + \gamma \frac{\partial E_t[R_{t+1}|A_t = a]}{\partial w_i} \right)$$

$$= \gamma E_t \left[ \frac{\partial}{\partial w_i} E_{t+1}[R_{t+1}] \right]$$

Where we are able to drop $\frac{\partial E_t[r_{t+1}|A_t = a]}{\partial w_i}$ because the immediate reward is independent of the state in memory once conditioned on the action. Thus we finally arrive at:

$$\frac{\partial}{\partial w_i} E_t[R_t] = \frac{1}{w_i} P_t(m_t = S_i) \left( \sum_{a \in \mathcal{A}} \pi(a|S_t, S_i) E_t[R_t|A_t = a] - E_t[R_t] \right)$$

$$+ \gamma E_t \left[ \frac{\partial}{\partial w_i} E_{t+1}[R_{t+1}] \right]$$

# B DERIVATION OF EXPRESSION FOR GRADIENT FOR MULTIPLE-STATE MEMORY

$$P_t(\mathcal{M}_t = S_{\hat{T}}) = \prod_{i \in \hat{T}} w_i \bigg/ \sum_{\tilde{T} \in \binom{T_t}{n}} \prod_{i \in \tilde{T}} w_i$$

$$E_t[R_t] = \sum_{\hat{T} \in \binom{T_t}{n}} P_t(\mathcal{M}_t = S_{\hat{T}}) \sum_{j \in \hat{T}} Q(S_j|S_{\hat{T}}) \sum_{a \in \mathcal{A}} \pi(a|S_t, S_j) E_t[R_t|A_t = a]$$

$$\frac{\partial}{\partial w_i} E_t[R_t] = \sum_{\hat{T} \in \binom{T_t}{n}} \left( \frac{\partial P_t(\mathcal{M}_t = S_{\hat{T}})}{\partial w_i} \sum_{j \in \hat{T}} Q(S_j|\mathcal{M}_t = S_{\hat{T}}) \sum_{a \in \mathcal{A}} \pi(a|S_t, S_j) E_t[R_t|A_t = a] \right.$$

$$\left. + P_t(\mathcal{M}_t = S_{\hat{T}}) \sum_{j \in \hat{T}} Q(S_j|S_{\hat{T}}) \sum_{a \in \mathcal{A}} \pi(a|S_t, S_j) \frac{\partial E_t[R_t|A_t = a]}{\partial w_i} \right)$$

Working out the first term:

$$\sum_{\hat{T} \in \binom{T_t}{n}} \frac{\partial P_t(\mathcal{M}_t = S_{\hat{T}})}{\partial w_i} \sum_{j \in \hat{T}} Q(S_j| = S_{\hat{T}}) \sum_{a \in \mathcal{A}} \pi(a|S_t, S_j) E_t[R_t|A_t = a]$$

$$= \sum_{\hat{T} \in \binom{T_t}{n}} P_t(\mathcal{M}_t = S_{\hat{T}}) \frac{\partial}{\partial w_i}(\log(P_t(\mathcal{M}_t = S_{\hat{T}}))) \sum_{j \in \hat{T}} Q(S_j|S_{\hat{T}})$$
$$\sum_{a \in \mathcal{A}} \pi(a|S_t, S_j) E_t[R_t|A_t = a]$$

$$= \sum_{\hat{T} \in (\binom{T_t}{n} \ni i)} \frac{1}{w_i} P_t(\mathcal{M}_t = S_{\hat{T}}) \sum_{j \in \hat{T}} Q(S_j|S_{\hat{T}}) \sum_{a \in \mathcal{A}} \pi(a|S_t, S_j) E_t[R_t|A_t = a]$$

$$- \sum_{\hat{T} \in \binom{T_t}{n}} P_t(\mathcal{M}_t = S_{\hat{T}}) \left( \frac{\sum_{\tilde{T} \in (\binom{T_t}{n} \ni i)} \prod_{j \in \tilde{T} \setminus \{i\}} w_j}{\sum_{\tilde{T} \in \binom{T_t}{n}} \prod_{j \in \tilde{T}} w_j} \right) \sum_{j \in \hat{T}} Q(S_j|S_{\hat{T}})$$
$$\sum_{a \in \mathcal{A}} \pi(a|S_t, S_j) E_t[R_t|A_t = a]$$

$$= \sum_{\hat{T} \in (\binom{T_t}{n} \ni i)} \left( \frac{\prod_{j \in \hat{T} \setminus \{i\}} w_j}{\sum_{\tilde{T} \in \binom{T_t}{n}} \prod_{j \in \tilde{T}} w_j} \right) \left( \sum_{j \in \hat{T}} Q(S_j|S_{\hat{T}}) \sum_{a \in \mathcal{A}} \pi(a|S_t, S_j) E_t[R_t|A_t = a] \right.$$
$$\left. - E_t[R_t] \right)$$

$$= \sum_{\hat{T} \in (\binom{T_t}{n} \ni i)} \frac{1}{w_i} P_t(\mathcal{M}_t = S_{\hat{T}}) \left( \sum_{j \in \hat{T}} Q(S_j|S_{\hat{T}}) \sum_{a \in \mathcal{A}} \pi(a|S_t, S_j) E_t[R_t|A_t = a] \right.$$
$$\left. - E_t[R_t] \right)$$

Working out the second term:

$$\sum_{\hat{T} \in \binom{T}{n}} P_t(\mathcal{M}_t = S_{\hat{T}}) \sum_{j \in \hat{T}} Q(S_j|S_{\hat{T}}) \sum_{a \in \mathcal{A}} \pi(a|S_t, S_j) \frac{\partial E_t[R_t|A_t = a]}{\partial w_i}$$

$$= \sum_{\hat{T} \in \binom{T}{n}} P_t(\mathcal{M}_t = S_{\hat{T}}) \sum_{j \in \hat{T}} Q(S_j|S_{\hat{T}}) \sum_{a \in \mathcal{A}} \pi(a|S_t, S_j) \left( \frac{\partial E_t[r_{t+1}|A_t = a]}{\partial w_i} \right.$$
$$\left. + \gamma \frac{\partial E_t[R_{t+1}|A_t = a]}{\partial w_i} \right)$$

$$= \gamma E_t \left[ \frac{\partial}{\partial w_i} E_{t+1}[R_{t+1}] \right]$$

So all together we get:

$$\frac{\partial}{\partial w_i} E_t[R_t] = \sum_{\hat{T} \in (\binom{T_t}{n} \ni i)} \frac{1}{w_i} P_t(\mathcal{M}_t = S_{\hat{T}}) \left( \sum_{j \in \hat{T}} Q(S_j|S_{\hat{T}}) \sum_{a \in \mathcal{A}} \pi(a|S_t, S_j) E_t[R_t|A_t = a] \right.$$
$$\left. - E_t[R_t] \right) + \gamma E_t \left[ \frac{\partial}{\partial w_i} E_{t+1}[R_{t+1}] \right]$$

## C    PROOF OF LEMMA 1

**Lemma 1.** *Selecting elements sequentially according to equation 12 will result in a vector $\hat{T}$ whose elements correspond to a sample drawn from equation 11.*

*Proof.* Selecting elements sequentially according to equation 12 gives the following probability for a given vector $\hat{T}$.

$$P(\hat{T}) = P(\hat{T}[0])P(\hat{T}[1]|\hat{T}[0])...P(\hat{T}[n-1]|\hat{T}[0:n-2])$$

$$= \frac{w_{\hat{T}[0]} \displaystyle\sum_{\tilde{T}\in\binom{T\backslash\hat{T}[0]}{n-1}} \prod_{j\in\tilde{T}} w_j}{n \displaystyle\sum_{\tilde{T}\in\binom{T}{n}} \prod_{j\in\tilde{T}} w_j} \cdot \frac{w_{\hat{T}[1]} \displaystyle\sum_{\tilde{T}\in\binom{T\backslash\hat{T}[0:1]}{n-2}} \prod_{j\in\tilde{T}} w_j}{(n-1) \displaystyle\sum_{\tilde{T}\in\binom{T\backslash\hat{T}[0]}{n-1}} \prod_{j\in\tilde{T}} w_j} \cdot ... \cdot \frac{w_{\hat{T}[n-1]}}{1 \displaystyle\sum_{\tilde{T}\in\binom{T\backslash\hat{T}[0:n-2]}{1}} \prod_{j\in\tilde{T}} w_j}$$

$$= \prod_{i\in\hat{T}} w_i \left/ \left( n! \sum_{\tilde{T}\in\binom{T}{n}} \prod_{i\in\tilde{T}} w_i \right) \right.$$

To complete the proof note that this is one of $n!$ vectors with the same set of elements in different order, each of which will have equal probability, hence to obtain the probability for the corresponding set we simply have to multiply by $n!$ which gives us equation 11. $\qquad\square$

## D    PROOF OF THEOREM 1

We divide the proof into a series of lemmas. First we will define some notation to facilitate referencing algorithm 1 in the proofs below. Let $T_t$ be the set $\{0, ..., t-1\}$ and $\hat{T}_t[0:i-1]$ be the value of $\hat{T}[0:i-1]$ at the call to UPDATE with parameter $t$. Note that $\hat{T}_t[0:-1] = \emptyset$. Let $\Omega_t[i]$ and $\tilde{\Omega}_t[i]$ be the values of $\Omega[i]$ and $\tilde{\Omega}[i]$ respectively, at the call to UPDATE with parameter $t$. Let $P_{t,i}$ be the value of P when it is set in loop index i of the loop starting at line 4 within the call to UPDATE with parameter t. Let $\Omega''_{t,i}$ and $\Omega'_{t,i}$ be the values of $\Omega''$ and $\Omega'$ respectively after they are set within index $i$ of the loop starting at line 4 within the UPDATE call with parameter $t$. Let $\omega_{t,i}$ and $\tau_{t,i}$ be the values of $\omega$ and $\tau$ respectively at the beginning of loop index i of the loop starting at line 4 within the call to UPDATE with parameter t. Note that $\omega_{t,i} = w_{\tau_{t,i}}$.

**Lemma 2.**

$$\left(T_{t+1} \backslash \hat{T}_{t+1}[0:i-1]\right) = \left(T_t \backslash \hat{T}_t[0:i-1]\right) \cup \{\tau_{t,i}\}$$

*Proof.* By design $T_{t+1} = T_t \cup \{\tau_{t,0}\}$. Towards a proof by induction assume that after choosing whether or not to swap $\hat{T}_t[i]$ we have:

$$\left(T_{t+1} \backslash \hat{T}_{t+1}[0:i-1]\right) = \left(T_t \backslash \hat{T}_t[0:i-1]\right) \cup \{\tau_{t,i}\}$$

Then on the next iteration either we swap $\hat{T}[i]$ for $\tau_{t,i}$ or we don't. If we do swap then $\tau_{t,i+1} = \hat{T}_t[i]$ and $\hat{T}_{t+1}[i] = \tau_{t,i}$ thus

$$\left(T_{t+1} \backslash \hat{T}_{t+1}[0:i]\right) = \left(T_{t+1} \backslash \hat{T}_{t+1}[0:i-1]\right) \backslash \{T_{t+1}[i]\}$$

$$= \left(T_t \backslash \hat{T}_t[0:i-1]\right) \cup \{\tau_{t,i}\} \backslash \{T_{t+1}[i]\}$$

$$= \left(T_t \backslash \hat{T}_t[0:i-1]\right) \cup \{\tau_{t,i}\} \backslash \{\tau_{t,i}\}$$

$$= \left(T_t \backslash \hat{T}_t[0:i-1]\right)$$

$$= \left(T_t \backslash \hat{T}_t[0:i]\right) \cup \{\hat{T}_t[i]\}$$

$$= \left(T_t \backslash \hat{T}_t[0:i]\right) \cup \{\tau_{t,i+1}\}$$

On the other hand if we do not swap then $\tau_{t,i+1} = \tau_{t,i}$ and $\hat{T}_{t+1}[i] = \hat{T}_t[i]$ thus

$$
\begin{aligned}
\left(T_{t+1} \setminus \hat{T}_{t+1}[0:i]\right) &= \left(T_{t+1} \setminus \hat{T}_{t+1}[0:i-1]\right) \setminus \{T_{t+1}[i]\} \\
&= \left(T_t \setminus \hat{T}_t[0:i-1]\right) \cup \{\tau_{t,i}\} \setminus \{T_{t+1}[i]\} \\
&= \left(T_t \setminus \hat{T}_t[0:i-1]\right) \cup \{\tau_{t+1,i}\} \setminus \{T_t[i]\} \\
&= \left(T_t \setminus \hat{T}_t[0:i]\right) \cup \{\tau_{t,i+1}\}
\end{aligned}
$$

Which suffices to complete the inductive proof. $\qquad\square$

**Lemma 3.**

$$
\sum_{\tilde{T} \in \binom{T \cup \{\hat{t}\}}{m}} \prod_{\tilde{t} \in \tilde{T}} w_{\tilde{t}} = \sum_{\tilde{T} \in \binom{T}{m}} \prod_{\tilde{t} \in \tilde{T}} w_{\tilde{t}} + w_{\hat{t}} \cdot \sum_{\tilde{T} \in \binom{T}{m-1}} \prod_{\tilde{t} \in \tilde{T}} w_{\tilde{t}}
$$

*Proof.* The proof follows from noting that the first sum on the right side includes every term of the left sum where $\tilde{T}$ does not contain $\hat{t}$, while the second term on the right side is equivalent to summing over those $\tilde{T}$ in the left sum that do contain $\hat{t}$. $\qquad\square$

**Lemma 4.** *For $t \geq n$:*

$$
\Omega_t[i] = \sum_{\tilde{T} \in \binom{T_t \setminus \hat{T}_t[0:i-1]}{n-i}} \prod_{\tilde{t} \in \tilde{T}} w_{\tilde{t}}, \ \forall i \in \{0, ..., n\}
$$

*and also*

$$
\tilde{\Omega}_t[i] = \sum_{\tilde{T} \in \binom{T_t \setminus \hat{T}_t[0:i-1]}{n-i-1}} \prod_{\tilde{t} \in \tilde{T}} w_{\tilde{t}}, \ \forall i \in \{0, ..., n-1\}
$$

*Proof.* Towards a proof by induction, assume the lemma holds at time $t$, a quick trace through algorithm 1 will show that the update leading to $\Omega_{t+1}[i]$ for $i \in \{0, ..., n-1\}$ is always:

$$
\Omega_{t+1}[i] = \Omega_t[i] + \omega_{t,i} \cdot \tilde{\Omega}_t[i]
$$

From here we can apply lemma 3, along with lemma 2 as follows to get the desired result:

$$
\begin{aligned}
\Omega_{t+1}[i] &= \Omega_t[i] + \omega_{t,i} \cdot \tilde{\Omega}_t[i] \\
&= \sum_{\tilde{T} \in \binom{T_t \setminus \hat{T}_t[0:i-1]}{n-i}} \prod_{\tilde{t} \in \tilde{T}} w_{\tilde{t}} + \omega_{t,i} \cdot \sum_{\tilde{T} \in \binom{T_t \setminus \hat{T}_t[0:i-1]}{n-i-1}} \prod_{\tilde{t} \in \tilde{T}} w_{\tilde{t}} \\
&= \sum_{\tilde{T} \in \binom{T_{t+1} \setminus \hat{T}_{t+1}[0:i-1]}{n-i}} \prod_{\tilde{t} \in \tilde{T}} w_{\tilde{t}}
\end{aligned}
$$

Now note that $\tilde{\Omega}_t[n] = 1$ for all $t$ which is also the correct value thus we have completed the induction step for the first half of the lemma.

On the other hand the update leading to $\tilde{\Omega}_{t+1}[i]$ is as follows:

$$
\tilde{\Omega}_{t+1}[i] = \Omega_{t+1}[i+1] + w_{\hat{T}_{t+1}[i]} \cdot \tilde{\Omega}_{t+1}[i+1]
$$

As a second level of induction assume the lemma holds for $\tilde{\Omega}_{t+1}[i+1]$ then applying lemma 3 we get:

$$
\begin{aligned}
\tilde{\Omega}_{t+1}[i] &= \Omega_{t+1}[i+1] + w_{\hat{T}_{t+1}[i]} \cdot \tilde{\Omega}_{t+1}[i+1] \\
&= \sum_{\tilde{T} \in \binom{T_{t+1} \setminus \hat{T}_{t+1}[0:i]}{n-i-1}} \prod_{\tilde{t} \in \tilde{T}} w_{\tilde{t}} + w_{\hat{T}_{t+1}[i]} \cdot \sum_{\tilde{T} \in \binom{T_{t+1} \setminus \hat{T}_{t+1}[0:i]}{n-i-2}} \prod_{\tilde{t} \in \tilde{T}} w_{\tilde{t}} \\
&= \sum_{\tilde{T} \in \binom{T_{t+1} \setminus \hat{T}_{t+1}[0:i-1]}{n-i-1}} \prod_{\tilde{t} \in \tilde{T}} w_{\tilde{t}}
\end{aligned}
$$

As the base case for this second level of induction, note that $\tilde{\Omega}_{t+1}[n-1] = 1$ for all $t$ which is also the correct value, thus assuming $\Omega_{t+1}$ is correct we will also get the correct values for $\tilde{\Omega}_{t+1}$. This completes the induction step for the lemma, it remains to prove the base case.

Note that $\Omega[n]$ and $\tilde{\Omega}[n-1]$ each start at 1. The beginning of the algorithm before line 19 is then intended to initialize all $\Omega$ and $\tilde{\Omega}$ values to have the correct value at time $t = n$, to see that this is the case, first note $\forall i \in \{0, ..., n\}$:

$$\Omega_n[i] = \prod_{j=i}^{n-1} w_{\hat{T}_n[j]}$$

$$= \sum_{\tilde{T} \in \binom{T_n \setminus \hat{T}_n[0:i-1]}{n-i}} \prod_{\tilde{t} \in \tilde{T}} w_{\tilde{t}}$$

And also, by induction on the trivial $i = n - 1$ case, $\forall i \in \{0, ..., n-2\}$:

$$\tilde{\Omega}_n[i] = \Omega_n[i+1] + w_{\hat{T}_n[i]} \cdot \tilde{\Omega}_n[i+1]$$

$$= \sum_{\tilde{T} \in \binom{T_n \setminus \hat{T}_n[0:i]}{n-i-1}} \prod_{\tilde{t} \in \tilde{T}} w_{\tilde{t}} + w_{\hat{T}_n[i]} \cdot \sum_{\tilde{T} \in \binom{T_n \setminus \hat{T}_n[0:i]}{n-i-2}} \prod_{\tilde{t} \in \tilde{T}} w_{\tilde{t}}$$

$$= \sum_{\tilde{T} \in \binom{T_n \setminus \hat{T}_n[0:i-1]}{n-i-1}} \prod_{\tilde{t} \in \tilde{T}} w_{\tilde{t}}$$

Thus indeed the lemma holds at time $n$ which serves as a base case for the rest of the proof. $\qquad\square$

**Lemma 5.** $P_{t,i} = 1 - \frac{\hat{P}(\hat{T}_t[i] | \hat{T}_{t+1}[0:i-1]; T_{t+1}, n)}{\hat{P}(\hat{T}_t[i] | \hat{T}_t[0:i-1]; T_t, n)}$

*Proof.* Substituting the definition from equation 12, along with the value assigned to $P_{t,i}$ and simplifying slightly what we wish to show is:

$$\frac{\Omega''_{t,i} \Omega_t[i]}{\Omega'_{t,i} \Omega_t[i+1]} = \frac{\sum_{\tilde{T} \in \binom{T_{t+1} \setminus (\hat{T}_{t+1}[0:i-1] \cup \{\hat{T}_t[i]\})}{n-i-1}} \prod_{j \in \tilde{T}} w_j \cdot \sum_{\tilde{T} \in \binom{T_t \setminus \hat{T}_t[0:i-1]}{n-i}} \prod_{j \in \tilde{T}} w_j}{\sum_{\tilde{T} \in \binom{T_{t+1} \setminus \hat{T}_{t+1}[0:i-1]}{n-i}} \prod_{j \in \tilde{T}} w_j \cdot \sum_{\tilde{T} \in \binom{T_t \setminus \hat{T}_t[0:i]}{n-i-1}} \prod_{j \in \tilde{T}} w_j}$$

Substituting in the values of $\Omega_t[i]$ and $\Omega_{t+1}[i]$ from lemma 4 into the above formula, it will suffice to apply lemma 3 and lemma 4 to additionally show:

$$\Omega'_{t,i} = \Omega_t[i] + \omega_{t,i} \cdot \tilde{\Omega}_t[i]$$

$$= \sum_{\tilde{T} \in \binom{T_t \setminus \hat{T}_t[0:i-1]}{n-i}} \prod_{\tilde{t} \in \tilde{T}} w_{\tilde{t}} + \omega_{t,i} \cdot \sum_{\tilde{T} \in \binom{T_t \setminus \hat{T}_t[0:i-1]}{n-i-1}} \prod_{\tilde{t} \in \tilde{T}} w_{\tilde{t}}$$

$$= \sum_{\tilde{T} \in \binom{T_{t+1} \setminus \hat{T}_{t+1}[0:i-1]}{n-i}} \prod_{j \in \tilde{T}} w_j$$

and for $0 \leq i \leq n - 2$

$$\Omega''_{t,i} = \Omega_t[i+1] + \omega_{t,i} \cdot \tilde{\Omega}_t[i+1]$$

$$= \sum_{\tilde{T} \in \binom{T_t \setminus \hat{T}_t[0:i]}{n-i-1}} \prod_{\tilde{t} \in \tilde{T}} w_{\tilde{t}} + \omega_{t,i} \cdot \sum_{\tilde{T} \in \binom{T_t \setminus \hat{T}_t[0:i]}{n-i-2}} \prod_{\tilde{t} \in \tilde{T}} w_{\tilde{t}}$$

$$= \sum_{\tilde{T} \in \binom{T_{t+1} \setminus (\hat{T}_{t+1}[0:i-1] \cup \{\hat{T}_t[i]\})}{n-i-1}} \prod_{j \in \tilde{T}} w_j$$

The first is a simple application of lemma 2. The second follows from a similar observation in addition to noting that $\hat{T}_t[i]$ has been removed from each term. For $i = n - 1$, $\Omega''_{t,i} = 1$ which trivially obeys the same formula. $\qquad\square$

**Lemma 6.** $P_{t,i} \geq 0$.

*Proof.*

$$P_{t,i} = 1 - \frac{\displaystyle\sum_{\tilde{T}\in\binom{T_{t+1}\setminus(\hat{T}_{t+1}[0:i-1]\cup\{\hat{T}_t[i]\})}{n-i-1}}\prod_{j\in\tilde{T}}w_j \sum_{\tilde{T}\in\binom{T_t\setminus\hat{T}_t[0:i-1]}{n-i}}\prod_{j\in\tilde{T}}w_j}{\displaystyle\sum_{\tilde{T}\in\binom{T_{t+1}\setminus\hat{T}_{t+1}[0:i-1]}{n-i}}\prod_{j\in\tilde{T}}w_j \sum_{\tilde{T}\in\binom{T_t\setminus\hat{T}_t[0:i]}{n-i-1}}\prod_{j\in\tilde{T}}w_j}$$

hence

$P_{t,i} \geq 0$

$$\Longleftrightarrow \sum_{\tilde{T}\in\binom{T_{t+1}\setminus\hat{T}_{t+1}[0:i-1]}{n-i}}\prod_{j\in\tilde{T}}w_j \sum_{\tilde{T}\in\binom{T_t\setminus\hat{T}_t[0:i]}{n-i-1}}\prod_{j\in\tilde{T}}w_j$$

$$\geq \sum_{\tilde{T}\in\binom{T_{t+1}\setminus(\hat{T}_{t+1}[0:i-1]\cup\{\hat{T}_t[i]\})}{n-i-1}}\prod_{j\in\tilde{T}}w_j \sum_{\tilde{T}\in\binom{T_t\setminus\hat{T}_t[0:i-1]}{n-i}}\prod_{j\in\tilde{T}}w_j$$

$$\overset{lem. 3}{\Longleftrightarrow} \left(\sum_{\tilde{T}\in\binom{T_{t+1}\setminus(\hat{T}_{t+1}[0:i-1]\cup\{\hat{T}_t[i]\})}{n-i}}\prod_{j\in\tilde{T}}w_j + w_{\hat{T}_t[i]}\sum_{\tilde{T}\in\binom{T_{t+1}\setminus(\hat{T}_{t+1}[0:i-1]\cup\{\hat{T}_t[i]\})}{n-i-1}}\prod_{j\in\tilde{T}}w_j\right)$$
$$\sum_{\tilde{T}\in\binom{T_t\setminus\hat{T}_t[0:i]}{n-i-1}}\prod_{j\in\tilde{T}}w_j$$

$$\geq \left(\sum_{\tilde{T}\in\binom{T_t\setminus\hat{T}_t[0:i]}{n-i}}\prod_{j\in\tilde{T}}w_j + w_{\hat{T}_t[i]}\sum_{\tilde{T}\in\binom{T_t\setminus\hat{T}_t[0:i]}{n-i-1}}\prod_{j\in\tilde{T}}w_j\right)\sum_{\tilde{T}\in\binom{T_{t+1}\setminus(\hat{T}_{t+1}[0:i-1]\cup\{\hat{T}_t[i]\})}{n-i-1}}\prod_{j\in\tilde{T}}w_j$$

$$\Longleftrightarrow \sum_{\tilde{T}\in\binom{T_{t+1}\setminus(\hat{T}_{t+1}[0:i-1]\cup\{\hat{T}_t[i]\})}{n-i}}\prod_{j\in\tilde{T}}w_j \sum_{\tilde{T}\in\binom{T_t\setminus\hat{T}_t[0:i]}{n-i-1}}\prod_{j\in\tilde{T}}w_j$$

$$\geq \sum_{\tilde{T}\in\binom{T_t\setminus\hat{T}_t[0:i]}{n-i}}\prod_{j\in\tilde{T}}w_j \sum_{\tilde{T}\in\binom{T_{t+1}\setminus(\hat{T}_{t+1}[0:i-1]\cup\{\hat{T}_t[i]\})}{n-i-1}}\prod_{j\in\tilde{T}}w_j$$

$$\overset{lem. 3}{\Longleftrightarrow} \left(\sum_{\tilde{T}\in\binom{T_t\setminus\hat{T}_t[0:i]}{n-i}}\prod_{\tilde{t}\in\tilde{T}}w_{\tilde{t}} + \omega_{t,i}\cdot\sum_{\tilde{T}\in\binom{T_t\setminus\hat{T}_t[0:i]}{n-i-1}}\prod_{\tilde{t}\in\tilde{T}}w_{\tilde{t}}\right)\sum_{\tilde{T}\in\binom{T_t\setminus\hat{T}_t[0:i]}{n-i-1}}\prod_{j\in\tilde{T}}w_j$$

$$\geq \left(\sum_{\tilde{T}\in\binom{T_t\setminus\hat{T}_t[0:i]}{n-i-1}}\prod_{\tilde{t}\in\tilde{T}}w_{\tilde{t}} + \omega_{t,i}\cdot\sum_{\tilde{T}\in\binom{T_t\setminus\hat{T}_t[0:i]}{n-i-2}}\prod_{\tilde{t}\in\tilde{T}}w_{\tilde{t}}\right)\sum_{\tilde{T}\in\binom{T_t\setminus\hat{T}_t[0:i]}{n-i}}\prod_{j\in\tilde{T}}w_j$$

$$\Longleftrightarrow \left(\sum_{\tilde{T}\in\binom{T_t\setminus\hat{T}_t[0:i]}{n-i-1}}\prod_{j\in\tilde{T}}w_j\right)^2 \geq \sum_{\tilde{T}\in\binom{T_t\setminus\hat{T}_t[0:i]}{n-i}}\prod_{j\in\tilde{T}}w_j \sum_{\tilde{T}\in\binom{T_t\setminus\hat{T}_t[0:i]}{n-i-2}}\prod_{j\in\tilde{T}}w_j$$

$$\Longleftrightarrow \sum_{\tilde{T}\in\binom{T_t\setminus\hat{T}_t[0:i-1]}{n-i-1}}\sum_{\tilde{T}'\in\binom{T_t\setminus\hat{T}_t[0:i-1]}{n-i-1}}\prod_{j\in\tilde{T}}w_j\prod_{j\in\tilde{T}'}w_j$$

$$\geq \sum_{\tilde{T}\in\binom{T_t\setminus\hat{T}_t[0:i-1]}{n-i-2}}\sum_{\tilde{T}'\in\binom{T_t\setminus\hat{T}_t[0:i-1]}{n-i}}\prod_{j\in\tilde{T}}w_j\prod_{j\in\tilde{T}'}w_j$$

It is relatively straightforward to show that the lemma holds in this last form. To do so note that except for terms for which $\tilde{T}$ and $\tilde{T}'$ are identical on the left (which are not possible on the right),

each term on the left of the inequality is also present on the right, however the number of repetitions of each term varies between the left and right. In the left sum if a term includes m values shared between $\tilde{T}$ and $\tilde{T}'$, this term will appear $\binom{2(n-i-1-m)}{n-i-1-m}$ times. This is because we can choose $n - i - m$ non-duplicate values to be in $\tilde{T}$ and place the rest in $\tilde{T}'$, each of these permutations will correspond to a term in the sum. On the other hand in the left sum if a term includes m values shared between $\tilde{T}$ and $\tilde{T}'$, this term will appear $\binom{2(n-i-1-m)}{n-i-2-m}$ times. Similarly this is because in this case we can choose $n - i - 1 - m$ non-duplicate values to be in $\tilde{T}$ and place the rest in $\tilde{T}'$, each of these permutations will correspond to a different term in the sum.

Since $\binom{2N}{N} > \binom{2N}{N-1}$, $\forall N$ every term which is present on the right side is present on the left with more repetitions and thus the left side must be greater than the right and the lemma holds. $\quad\square$

This lemma shows that $P$ is indeed a valid probability, which means that swapping according to it in algorithm 1 is admissible.

**Theorem 1.** $\forall t \geq n,\ 0 \leq i \leq n - 1$:

$$P(\hat{T}_t[i] = t_i | \hat{T}_t[0 : i - 1] = [t_0, ..., t_{i-1}]) = \hat{P}(t_i | [t_0, ..., t_{i-1}]; T_t, n)$$

where $[t_0, ..., t_i]$ is an arbitrary vector of unique elements of $T_t$.

*Proof.* Let $\hat{T}_{i,t}$ be the value of $\hat{T}$ right before making the swap decision on line 12 in loop index i of the loop starting at line 4 within the call to UPDATE with parameter t. Towards a proof by induction assume that at time index $t$ directly prior to making the swap decision for $\hat{T}[i]$ at line 12 of UPDATE we have for any arbitrary vector $[t_0, ..., t_{i-1}]$ of unique elements of $T_t$:

$$P(\hat{T}_{i,t}[j] = t_j | \hat{T}_t[0 : j - 1] = [t_0, ..., t_{j-1}]) = \hat{P}(t_j | [t_0, ..., t_{j-1}]; T_t, n),$$
$$\forall j \ s.t.\ i \leq j \leq n - 1$$

and for any arbitrary vector $[t'_0, ..., t'_{i-1}]$ of unique elements of $T_{t+1}$:

$$P(\hat{T}_{i,t}[j] = t'_j | \hat{T}_{t+1}[0 : j - 1] = [t'_0, ..., t'_{j-1}]) = \hat{P}(t_j | [t'_0, ..., t'_{j-1}]; T_{t+1}, n),$$
$$\forall j \ s.t.\ 0 \leq j \leq i - 1$$

That is all elements of $\hat{T}$ from 0 to $i - 1$ have the desired probability conditioned on proceeding elements of $\hat{T}_{t+1}$ while all elements from $i$ to $n - 1$ still have the desired probability when conditioned on proceeding elements of $\hat{T}_t$. This is a natural inductive assumption given we have already made our swap decisions up to but not including $i$ and are just about to make our decision for $i$.

Consider two mutually possible selections of $\hat{T}_{t+1}[0 : i - 1] = [t'_0, ..., t'_{i-1}]$ and $\hat{T}_t[0 : i - 1] = [t_0, ..., t_{i-1}]$ and note that together these uniquely determine the value $\tau_{t,i}$. Fixing these values, the only possible way to end up with $\hat{T}_{t+1}[i] = \hat{t}$ for $\hat{t} \in T_t \setminus \{t_0, ..., t_{j-1}\}$ is to have $\hat{T}_t[i] = \hat{t}$ and then choose not to swap $\hat{T}[i]$ at time $t$. Thus, substituting in $1 - P_{t,i}$ using the expression for $P_{t,i}$ obtained in lemma 5 we get:

$$P(\hat{T}_{t+1}[i] = \hat{t} | \hat{T}_{t+1}[0 : i - 1] = [t'_0, ..., t'_{j-1}], \hat{T}_t[0 : i - 1] = [t_0, ..., t_{j-1}])$$
$$= \hat{P}(\hat{t} | [t_0, ..., t_{i-1}]; T_t, n) \frac{\hat{P}(\hat{t} | [t'_0, ..., t'_{i-1}]; T_{t+1}, n)}{\hat{P}(\hat{t} | [t_0, ..., t_{i-1}]; T_t, n)}$$
$$= \hat{P}(\hat{t} | [t'_0, ..., t'_{i-1}]; T_{t+1}, n)$$

On the other hand to end up with $\hat{T}_{t+1}[i] = \tau_{t,i}$ we may start with any $\hat{t} \in T_t \setminus (\{t_0, ..., t_{j-1}\})$ and then choose to swap $\hat{T}[i]$ at time $t$, in this case we get:

$$P(\hat{T}_{t+1}[i] = \tau_{t,i}|\hat{T}_{t+1}[0:i-1] = [t'_0, ..., t'_{j-1}], \hat{T}_t[0:i-1] = [t_0, ..., t_{j-1}])$$

$$= \sum_{\hat{t} \in T_t \setminus \{t_0, ..., t_{j-1}\}} \hat{P}(\hat{t}|[t_0, ..., t_{j-1}]; T_t, n) \left(1 - \frac{\hat{P}(\hat{t}|[t'_0, ..., t'_{j-1}]; T_{t+1}, n)}{\hat{P}(\hat{t}|[t_0, ..., t_{j-1}]; T_t, n)}\right)$$

$$= \sum_{\hat{t} \in T_t \setminus \{t_0, ..., t_{j-1}\}} \hat{P}(\hat{t}|[t_0, ..., t_{j-1}]; T_t, n) - \hat{P}(\hat{t}|[t'_0, ..., t'_{j-1}]; T_{t+1}, n)$$

$$= 1 - \sum_{\hat{t} \in T_t \setminus \{t_0, ..., t_{j-1}\}} \hat{P}(\hat{t}|[t'_0, ..., t'_{j-1}]; T_{t+1}, n)$$

$$= 1 - \sum_{\hat{t} \in T_{t+1} \setminus (\{t'_0, ..., t'_{j-1}\} \cup \{\tau_{t,i}\})} \hat{P}(\hat{t}|[t'_0, ..., t'_{j-1}]; T_{t+1}, n)$$

$$= \hat{P}(\tau_{t,i}|[t'_0, ..., t'_{j-1}]; T_{t+1}, n)$$

Thus for any fixed mutually possible selections of $\hat{T}_{t+1}[0:i-1] = [t'_0, ..., t'_{j-1}]$ and $\hat{T}_t[0:i-1] = [t_0, ..., t_{j-1}]$ for both $\tau_{t,i}$ and all other $\hat{t} \in T_{t+1} \setminus \{t'_0, ..., t'_{j-1}\}$ we end up with correct conditional probabilities for $\hat{T}_{t+1}[i]$ assuming they are correct for $\hat{T}_t[i]$. Now note that the resulting probabilities are ultimately independent of $[t_0, ..., t_{j-1}]$, hence we would get the same values by conditioning on $[t'_0, ..., t'_{j-1}]$ alone. Thus if our inductive assumption holds we have for any arbitrary vector $[t'_0, ..., t'_{i-1}]$ of unique elements of $T_{t+1}$:

$$P(\hat{T}_{i+1,t}[j] = t'_j|\hat{T}_{t+1}[0:j-1] = [t'_0, ..., t'_{j-1}]) = \hat{P}(t_j|[t'_0, ..., t'_{j-1}]; T_{t+1}, n),$$
$$\forall j \; s.t. \; 0 \leq j \leq i$$

Which completes the inductive step. To prove the base case note that we initialize $\hat{T}$ such that at time $n$ it is filled with all available items in random order. It is easy to show that equation 12 implies that the probability of any ordering of a given set $\hat{T}$ is equal thus if the items in $T$ exactly fill $\hat{T}$ then ordering them at random will give the desired probabilities $\hat{P}(\hat{T}_n[i]|\hat{T}_{t+1}[0:i-1]; T_{t+1}, n)$, $\forall i$ s.t. $0 \leq i \leq n-1$, which gives us the base case to complete the inductive proof. □

# E LENGTH 20 ENVIRONMENT EXPERIMENT

Figure 6 shows the results of an experiment on the secret informant problem with environment length 20. Comparing figures 5 and 6 the number of episodes to convergence increases from $50,000$ to around $80,000$ with a doubling of the environment length while training still remains stable.

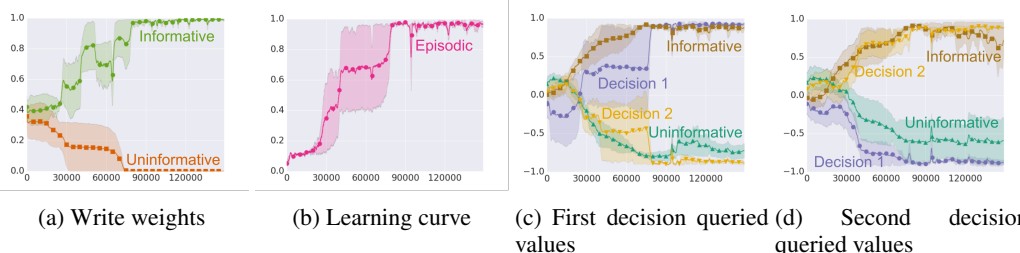

(a) Write weights  (b) Learning curve  (c) First decision queried values  (d) Second decision queried values

Figure 6: Experiment with environment length 20, 2 decisions and a 3 state memory. (a) shows average write weight assigned to informative states (●) and uninformative states (■). (b) shows average return with episodic memory learner (●), we did not train a recurrent baseline for this problem. (c) shows the value of several relevant query vector elements in the decision state. ▲ corresponds to the uninformative indicator, ■ to the informative indicator, ● to the first decision state identifier, ▼ to the second decision state identifier. (d) shows the same thing but for the query generated in the second decision state.

