# OpenReview forum: "Integrating Episodic Memory into a Reinforcement Learning Agent Using Reservoir Sampling"
_ICLR.cc/2018/Conference — Reject_

### Official Review · AnonReviewer2 · 2017-11-30
**A new version of episodic memory for DRL with less convincing results**

**Rating:** 4
**Confidence:** 4

**Review:**

This paper considers a new way to incorporate episodic memory with shallow-neural-nets RL using reservoir sampling. The authors propose a reservoir sampling algorithm for drawing samples from the memory. Some theoretical guarantees for the efficiency of reservoir sampling are provided. The whole algorithm is tested on a toy problem with 3 repeats. The comparisons between this episodic approach and recurrent neural net with basic GRU memory show the advantage of proposed algorithm.

The paper is well written and easy to understand. Typos didn't influence reading. It is a novel setup to consider reservoir sampling for episodic memory. The theory part focuses on effectiveness of drawing samples from the reservoir. Physical meanings of Theorem 1 are not well represented. What are the theoretical advantages of using reservoir sampling?

Four simple, shallow neural nets are built as query, write, value, and policy networks. The proposed architecture is only compared with a recurrent baseline with 10-unit GRU network. It is not clear the better performance comes from reservoir sampling or other differences. Moreover, the hyperparameters are not optimized on different architectures. It is hard to justify the empirically better performance without hyperparameter tuning. The authors mentioned that the experiments are done on a toy problem, only three repeats for each experiment. The technically soundness of this work is weakened by the experiments.

---

> ### Author Response · Authors · 2017-12-06
> **The meaning of theorem 1 and the advantage of reservoir sampling**
>
> Thank you for your feedback, points about the experimental design are discussed in the general comments.
>
> Re:
> Physical meanings of Theorem 1 are not well represented. What are the theoretical advantages of using reservoir sampling?
>
> This is partially addressed in the general comment at the top. The main advantage of using reservoir sampling is computationally. Naively drawing a weighted sample of n states from the entire history of visited states would require us to store this entire history to draw from. Reservoir sampling allows us to update a sample online (in O(n) time in the size of the memory) while avoiding storing any states not included in the current sample. Because we are able to do this we can then use a policy gradient like approach to train the memory to preferentially store states that lead to greater advantage when recalled.

---

### Official Review · AnonReviewer1 · 2017-11-30
**Interesting new ideas - more work needed to justify approach**

**Rating:** 4
**Confidence:** 3

**Review:**

The paper proposes a modified approach to RL, where an additional "episodic memory" is kept by the agent. What this means is that the agent has a reservoir of n "states" in which states encountered in the past can be stored. There are then of course two main questions to address (i) which states should be stored and how (ii) how to make use of the episodic memory when deciding what action to take.

For the latter question, the authors propose using a "query network" that based on the current state, pulls out one state from the memory according to certain probability distribution. This network has many tunable parameters, but the main point is that the policy then can condition on this state drawn from the memory. Intuitively, one can see why this may be advantageous as one gets some information from the past. (As an aside, the authors of course acknowledge that recurrent neural networks have been used for this purpose with varying degrees of success.)

The first question, had a quite an interesting and cute answer. There is a (non-negative) importance weight associated with each state and a collection of states has weight that is simply the product of the weights. The authors claim (with some degree of mathematical backing) that sampling a memory of n states where the distribution over the subsets of past states of size n is proportional to the product of the weights is desired. And they give a cute online algorithm for this purpose. However, the weights themselves are given by a network and so weights may change (even for states that have been observed in the past). There is no easy way to fix this and for the purpose of sampling the paper simply treats the weights as immutable.

There is also a toy example created to show that this approach works well compared to the RNN based approaches.

Positives:

- An interesting new idea that has potential to be useful in RL
- An elegant algorithm to solve at least part of the problem properly (the rest of course relies on standard SGD methods to train the various networks)

Negatives:
- The math is fudged around quite a bit with approximations that are not always justified
- While overall the writing is clear, in some places I feel it could be improved. I had a very hard time understanding the set-up of the problem in Figure 2. [In general, I also recommend against using figure captions to describe the setup.]
- The experiments only demonstrate the superiority of this method on an example chosen artificially to work well with this approach.

---

> ### Author Response · Authors · 2017-12-06
> **On the various approximations used**
>
> Thank you for the detailed feedback, these are some very helpful points.
>
> Re:
> However, the weights themselves are given by a network and so weights may change (even for states that have been observed in the past).
>
> This is correct however it should be noted that this problem does not exist if we were to use batch training, i.e. running full episodes and training on the total resulting loss at the end. The issue is then similar to issues that arise when training recurrent models in an online setting (e.g. present recurrent state is the result of now stale past parameters). Though we do not prove it in this work we conjecture that this will not be an issue in the limit of small step sizes.
>
> Re:
> -The math is fudged around quite a bit with approximations that are not always justified
>
> While we tried to be as explicit as possible about where approximations were used, there are many aspects we are working to improve in the future. To give one example: how to take advantage of the interaction between the query and write network to better justify training only the write weight of the queried item and not every item presently in memory seems like a fruitful area for further theoretical investigation and improvement.
>
> Re:
> -While overall the writing is clear, in some places I feel it could be improved. I had a very hard time understanding the set-up of the problem in Figure 2. [In general, I also recommend against using figure captions to describe the setup.]
>
> Thank you for pointing this out, we agree the task could be explained more clearly and will work on revising this.

---

### Official Review · AnonReviewer3 · 2017-12-04
**A new RL model with episodic memory - robust experiments needed to justify the conclusion**

**Rating:** 4
**Confidence:** 3

**Review:**

This paper proposes one RL architecture using external memory for previous states, with the purpose of solving the non-markov tasks. The essential problems here are how to identify which states should be stored and how to retrieve memory during action prediction. The proposed architecture could identify the ‘key’ states through assigning higher weights for important states, and applied reservoir sampling to control write and read on memory. The weight assigning (write) network is optimized for maximize the expected rewards. This article focuses on the calculation of gradient for write network, and provides some mathematical clues for that.

This article compares their proposed architecture with RNN (GRU with 10 hidden unit) in few toy tasks. They demonstrate that proposed model could work better and rational of write network could be observed. However, it seems that hyper-parameters for RNN haven’t been tuned enough. It is because the toy task author demonstrates is actually quite similar to copy tasks, that previous state should be remembered. To my knowledge, copy task could be solved easily for super long sequence through RNN model. Therefore, empirically, it is really hard to justify whether this proposed method could work better. Also, intuitively, this episodic memory method should work better on long-term dependencies task, while this article only shows the task with 10 timesteps.

According to that, the experiments they demonstrated in this article are not well designed so that the conclusion they made in this article is not robust enough.

---

> ### Author Response · Authors · 2017-12-06
> **Comparing our task with the copy task**
>
> Thank you for your feedback, we agree that more work is needed to evaluate the empirical usefulness of our approach, our aim in this work was simply to demonstrably produce an external memory mechanism for which we could estimate gradients for training without back-propagation through time. We have explained our methodology in more detail in the general comment above.
>
> Re:
> It is because the toy task author demonstrates is actually quite similar to copy tasks, that previous state should be remembered.
>
> In the prior work you mention which trains RNNs on a copy task it is likely they were working in a sequence prediction framework rather than reinforcement learning. In sequence prediction the agent would process characters one at a time while outputting a probability distribution for each next character. The training would then proceed by increasing the probability of the correct characters as they are revealed providing a dense signal for improvement. The reinforcement learning framework assumes much less structure in both the problem and the provided feedback than this. The agent is not provided with the correct action at each step as a learning signal, it must instead learn this through trial and error, and rewards for the correct action are not necessarily given immediately but may be delayed as is the case in our problem (particularly in the case of more than one decision state). To train on this task in a manner similar to sequence prediction we would have to assume significantly more structure than the conventional reinforcement learning framework. These factors make the problem significantly more difficult, although it may be superficially similar.
>
> Re:
> Also, intuitively, this episodic memory method should work better on long-term dependencies task, while this article only shows the task with 10 time-steps.
>
> Appendix E shows a version of the task with 20 time-steps which gives an early indication that it scales quite well with length, we agree scaling to much longer tasks would be informative to explore in the future.

---

### Author Response · Authors · 2017-12-06
**1: Motivation of our approach as online version of external memory for RL. 2: Reasons for experimenting on a simple toy problem.**

The authors thank the reviewers for their helpful feedback.

One thing we would like to clarify is that the primary aim of this paper is not to propose a method to compete with recurrent neural networks (or their many variants) for reinforcement learning. Rather we build on the body of work which makes use of neural networks acting on external memories which has been shown to be useful in many cases. However, existing work involving external memory in a reinforcement learning or sequence prediction setting usually uses back-propagation through time for training. This requires computation proportional to the history length which is not acceptable in online reinforcement learning, particularly in the continuing learning setting where the history length grows arbitrarily long. If a model uses only recurrent computation this can be remedied to an extent by truncated back-propagation through time, at the cost of biasing the gradients and limiting the effective horizon of the model.

When using models which read from and write to external memories however, much of the apparent benefit comes from the ability to easily store information on much longer time scales. In this case it is less clear whether truncating gradients is a viable option and it makes sense to look into other gradient based methods which could work online with an external memory. The main contribution of this work is to present a particular framework for external memory in RL which, through the use of reservoir sampling, can be trained online (in O(n) time in the memory size) with approximate gradients without the need to perform back-propagation through time. For this reason we did not see it as meaningful or worthwhile at this point to compare with the many works which utilize external memories, or similarly temporal attention mechanisms, trained with back-propagation through time. We will make revisions to our introduction along these lines to make the motivation and intended use-case of our algorithm more clear.

The environment used in the experiments is intentionally simple to allow straightforward analysis and intuitive understanding of what the agent is doing. This is exemplified by the included plots of write weights and queried values which demonstrate that the individual components of our agent are performing as expected. Similar plots would likely not be possible to obtain in a more complex environment. In general we agree with the reviewers that more robust experiments are needed to demonstrate the practical utility of this approach. We see this work as building a conceptual foundation for future work on more realistic environments.

The RNN baseline provided was in no way meant to be representative of state of the art on problems like the one we explore here, it was only intended to provide context for our main experimental result. The main result being that our external memory mechanism, trained online with no back propagation through time can in fact learn to remember the important states (which for purpose of illustration are known to us ahead of time in this case) in a reinforcement learning problem. With that said we acknowledge that performing a more thorough hyper-parameter sweep would make for a more meaningful baseline, thus we plan do this and add it in a revision once complete.

We will address remaining concerns to the individual reviews.

We thank the reviewers again for constructive feedback, and hope that our responses address your concerns.

---

### Decision · Program_Chairs · 2018-01-29
**ICLR 2018 Conference Acceptance Decision**

**Decision:**

Reject

**Comment:**

This paper presents a memory architecture for RL based on reservoir sampling, and is meant to be an alternative to RNNs. The reviewers consider the idea to be potentially interesting and useful, but have concerns about the mathematical justification. They also point out limitations in the experiments: in particular, use of artificial toy problems, and a lack of strong baselines. I don't think the paper is ready for ICLR publication in its current form.